# Pharmacist-led rapid uptitration clinic in heart failure patients with reduced ejection fraction: Our experience within a virtual ward

Hussein Alhakem[1]*, Angela Murphy[1], Liuba Fusco[1], Grant McQueen[1], Sarah Pearse[2], Jodian Barrett[2], Deirdre Linnard[1], Sadia Khan[1,3]

1 Chelsea & Westminster NHS Foundation Trust, London, England, 2 Northwest London Virtual Hospital, Imperial College Healthcare NHS Trust, London, England, 3 National Heart and Lung Institute, Imperial College London, London, England

* hussein.alhakem@nhs.net

## Abstract

Heart failure with reduced ejection fraction is a chronic, progressive medical condition affecting millions of individuals worldwide. It is associated with high morbidity and mortality. The use of "foundational quadruple therapy" titrated to the maximum tolerated doses improves survival, quality of life, and reduces heart failure-related hospitalisation. Despite this evidence, there is a consistent trend of suboptimal dose up-titration, prolonged optimisation periods, and early therapy discontinuation. Virtual wards offer a potential innovative solution in transforming heart failure management by combining rapid medication optimisation with remote monitoring to improve patient outcomes. This retrospective study employed a single-group pre-post design to evaluate the effectiveness of a prescribing pharmacist in the rapid uptitration of Guidelines Directed Medical Therapy (GDMT) in patients with heart failure with reduced ejection fraction within a virtual ward setting. The study assessed clinical outcomes of 86 patients at baseline, following discharge from the virtual ward (typically after 4 weeks), and at 3–6 months post-discharge. Improvements were seen in NYHA scores, cardiac systolic function, and Optimal Medical Therapy (OMT) scores. The median Left Ventricular Ejection Fraction increased from 29% at baseline to 39% post-optimisation, signifying improved myocardial performance and a reduction in the severity of left ventricular dysfunction. Post-optimisation, 37% of patients attained an optimal OMT score of 8, 52% attained an acceptable score (5–7), and only 5% remained in the suboptimal range (0–4). Additionally, 84% of patients were prescribed all four foundational therapies. There was no notable increase in adverse events such as hypotension, bradycardia, or hyperkalaemia. Remote up-titration of heart failure medications within a virtual ward environment is a promising approach, offering a fast, feasible, safe, and efficient treatment solution for patients who are otherwise undertreated.

**Data availability statement:** There are no restrictions to data sharing for this study, as all the data has been anonymised. The anonymised dataset supporting the findings of this study has been provided as part of the supporting material and is available in the file titled S1 Appendix.

**Funding:** The author(s) received no specific funding for this work.

**Competing interests:** The authors have declared that no competing interests exist.

## Author summary

Heart failure with reduced ejection fraction (HFrEF) occurs when the heart is unable to pump blood efficiently throughout the body, leading to symptoms like fatigue, shortness of breath, and fluid retention. Clinical guidelines world-wide recommend starting medical therapy as early as possible, adjusted to the highest tolerated doses, to avoid worsening of symptoms and emergency hospitalisation. However, this process requires frequent clinician reviews and close monitoring, making it time-consuming for both patients and healthcare providers thus leading to suboptimal doses and early discontinuation of therapy. The STRONG-HF study demonstrated that accelerated dose adjustments, paired with regular follow-up, not only improved heart failure symptoms and quality of life of patients but also significantly reduced hospital admission and death, highlighting the knowledge gap in treatment optimisation strategies. Virtual wards offer an innovative solution allowing patients to be monitored remotely at home while still receiving the necessary dose adjustments and clinical care. In this paper, we share our experience in implementing this pathway, which has allowed us to start treatment early and seamlessly adjust medication doses in accordance with patient-specific parameters leading to an improvement in heart function, New York Heart Association (NYHA) score, and a reduction in heart failure related hospitalisation.

## Introduction

Heart failure with reduced ejection fraction (HFrEF) is a chronic progressive medical condition affecting millions of individuals worldwide which is associated with high morbidity and mortality [1]. Its incidence is increasing due to an aging population and the growing incidence of risk factors such as obesity and diabetes [2,3]. The use of "foundational quadruple therapy", which includes betablockers, angiotensin-converting enzyme inhibitors (ACEIs)/angiotensin II receptor blockers (ARBs)/ angiotensin receptor-neprilysin inhibitors (ARNIs), mineralocorticoid receptor antagonists (MRAs), and sodium-glucose cotransporter 2 inhibitors (SGLT2is), titrated to the maximum tolerated doses has been shown to improve survival, quality of life, reduce the risk of hospitalisations, and re-admission rates [4].

Despite the recommendation for Guideline-Directed Medical Therapy (GDMT), data shows consistent trend of suboptimal dose up-titration, prolonged optimisation period, and early therapy discontinuation [5–8]. One multinational observational study reviewed target dose achievement of GDMT over a 12-month period in Sweden, UK and US between 2016–2019 and found ACEIs/ARBs/ARNIs and beta-blockers were infrequently titrated, 18% and 12%, respectively, whilst mineralocorticoid receptor antagonists were not often titrated [5].

Furthermore, It is estimated to take around 6–12 months to achieve target doses of GDMT, even if conducted as part of a structured intervention program that include

close monitoring and clinician-led support [9]. In fact, the stepwise approach of initiating and optimising foundational quadruple therapy in current practice often requires frequent clinical reviews and laboratory tests, with intervals ranging from 2-8 weeks between each dose titration [9].

Importantly, under-dosing of GDMT during the initial period post diagnosis of HFrEF negatively affects left ventricular reverse re-modelling [10]. The STRONG-HF study demonstrated that an intensive treatment strategy of rapid up-titration of HF medications with close follow-up was readily accepted by patients, improved HF symptoms, quality of life, and reduced the risk of 180-day all-cause death or heart failure readmission compared with usual care [8]. The aim of this study is to investigate the effectiveness of a prescribing pharmacist in the rapid uptitration of GDMT in patients with HFrEF, using The Optimal Medical Therapy (OMT) scoring tool. This tool was developed by the Heart Failure Academic Research Consortium (HFARC) to provide a standardised approach towards evaluating the effectiveness of GDMT optimisation [11]. Higher OMT scores indicate better adherence to GDMT, which is associated with improved outcomes, reduced hospitalisation, and mortality [11].

In particular, we compared OMT scores pre- and post-intervention within the heart failure virtual ward of Chelsea and Westminster NHS Foundation Trust.

The current project aligns with the Darzi report recommendations by utilising the virtual ward environment to promote improved patient outcomes and reduce heart failure related admissions.

## Method

### Study design

This retrospective study employed a single-group according to a base-post design to evaluate the effectiveness of a prescribing pharmacist in the rapid uptitration of GDMT in patients with HFrEF within a virtual ward setting. The study assessed clinical outcomes of 86 patients at baseline, following discharge from the virtual ward (typically after 4 weeks), and at 3–6 months post-discharge. The dataset used in this study, including raw values, is available in Supplementary S1 Appendix.

### Study population

Patients were identified following a new diagnosis of HFrEF, confirmed by a cardiology specialist on echocardiogram or cardiovascular magnetic resonance (CMR) scan, or following referral from a member of the cardiology multidisciplinary team in patients with a pre-existing diagnosis of HFrEF requiring medical optimisation.

**Inclusion criteria:**

- Adults (≥18 years).

- Diagnosis of HFrEF.

- Patients requiring initiation and optimisation of GDMT.

- Haemodynamically stable patients not requiring in-patient care.

- Patient/carer capable of speaking/understanding English.

- Consented, willing, able and suitably supported to receive treatment and monitoring at home.

**Exclusion criteria:**

- Patients already on target doses of GDMT.

- Patients with cognitive impairments that may limit medication adherence or ability to escalate concerns.

- Patients with contraindications to GDMT.

- Patients declining offer to enrol in rapid uptitration program.
- Patients using pharmacy prepared medication compliance aids (e.g., dosette boxes) due to the frequent medication changes during the uptitration period.

### Intervention

This involved initiating and rapidly up-titrating GDMT to the maximum tolerated doses as outlined in the latest heart failure guidelines. This included beta-blockers, ACEIs/ARBs/ARNIs, MRAs and SGLTs inhibitors. The up-titration criteria for beta-blockers and ACEIs/ARBs/ARNIs were in line with the STRONG-HF trial protocol, with an aim of achieving a 25% dose increase per week, over a 4-week period. Dose titrations were guided by patient symptoms, vital signs (HR $\geq$ 55 and SBP $\geq$ 95) and satisfactory laboratory data ($K^+ \leq 5.0$ mmol/L and eGFR $\geq$ 30mL/min/1.73m$^2$). The pharmacist's role also included educating patients on their heart failure diagnosis, medication counselling, and overseeing the rapid uptitration process.

### Communication

Upon admission to the virtual ward, patients were verbally counselled on their heart failure condition and the medications prescribed for its management. Digital written resources was also provided to support self-management of heart failure. General practitioners (GPs) were notified of the patient's admission to the virtual ward through written communication, which outlined the reason for admission, i.e., new diagnosis of heart failure, the medication titration protocol, and the counselling points discussed with the patient regarding their heart failure diagnosis. On discharge from the virtual ward, arrangements were made to ensure specialist cardiology and community heart failure follow-up after the rapid uptitration period.

### North West London Virtual Hospital (NWLVH)

The NWLVH is overseen by experienced practitioners from nursing and allied health professional (AHP) backgrounds. Its primary function is to review patient-submitted measurements and symptom data on a daily basis. Practitioners at the NWLVH contact patients during the initial monitoring period, if data submissions are missed, or when a pre-defined clinical escalation parameter is triggered. The NWLVH operates 7 days a week from 8:00 AM to 8:00 PM, providing support and timely intervention as needed. Outside of these hours, patients are advised to contact NHS 111 or emergency services (999) if they feel unwell or require urgent assistance.

### Technology and monitoring

Patients meeting the pathway criteria were onboarded onto the heart failure virtual ward with a set of agreed-upon clinical escalation parameters. A remote monitoring kit was provided to facilitate the daily recording of HR, BP and weight and to enable the reporting of qualitative data relating to heart failure symptom control and potential side effects to treatment. Readings submitted by patients were reviewed daily by the NWLVH and escalations were managed in accordance with the escalation pathway outlined in the HF Virtual Ward Standard Operating Procedure. The prescribing pharmacist virtually reviewed patients on a weekly basis to optimise therapy, assess symptom control, and to identify potential side effects to treatment. In addition, a multidisciplinary team (MDT) meeting was held weekly with the NWLVH to review all patients on the virtual ward, and to set an estimated date for discharge.

### Blood tests

Baseline blood testing included NT-proBNP, renal function test, thyroid function test, liver function test, lipid profile, glycosylated haemoglobin (HbA1c), full blood count, iron studies, and vitamin D3 levels. Follow-up testing of BNP, renal and liver function tests occurred at 2 weeks and again at the end of the up-titration period.

## Outcomes

The study evaluated both primary and secondary outcomes.

### Primary outcomes

- The proportion of patients reaching maximum-tolerated GDMT doses, measured using the OMT scores (Table 1).

- Improvement in left ventricular ejection fraction (LVEF).

- Improvement in NYHA score.

### Secondary outcomes

- Reduction in hospitalisations related to heart failure over the 3–6-month follow-up period.

- Adverse events, such as hypotension and hyperkalemia, during the uptitration period.

## Data collection

Data was collected at three time points:

- **Baseline**: Upon onboarding to the virtual ward.

- **Discharge from the virtual ward**: Approximately 4 weeks after the initial intervention.

- **3–6-month follow-up**: To assess longer-term outcomes such as change in LVEF and rehospitalisation.

## Statistical analysis

The collected dataset of multiple clinically measurable variables was analysed using the statistical add-on function of Excel and using ad-hoc Python open-source codes. First of all, the Kolmogorov-Smirnov (K-S) test was used to assess the normality of the left ventricular ejection fraction (LVEF) distribution. Once the test was applied to baseline LVEF yielded a K-S statistic of 0.1511 with a p-value of 0.0734. At a 5% significance level ($\alpha = 0.05$), since $p = 0.0734 > 0.05$, we failed to reject the null hypothesis, indicating that the LVEF values follow a Gaussian distribution. Consequently, we initially employed the z critical test for analysis on this type of data.

**Table 1. Optimal Medical Therapy score in patients with HFrEF.**

| Score | 0 | 1 | 2 |
|---|---|---|---|
| ACEi, ARB, ARNI (licensed in heart failure) | None | <50% max dose ACEi/ARB | • ≥50% max dose ACEi/ARB<br>• Any dose ARNI<br>• <50% max dose with intolerance (symptomatic hypotension, K+>5.5mmol/L, eGFR reduction in ≥30%) with higher doses |
| Beta-blocker (licensed in heart failure) | None | <50% max dose | • ≥50% max dose<br>• <50% max dose with resting HR<55bpm<br>• <50% max dose with intolerance (symptomatic hypotension or bradycardia) with higher doses |
| MRA | None | <50% max dose | • ≥50% max dose<br>• <50% max dose with K+>5.5mmol/L<br>• <50% max dose with intolerance (symptomatic hypotension, eGFR reduction in ≥30%) with higher doses |
| SGLT2i | None | – | • Any dose<br>• Documented intolerance or contra-indication |

Scoring: 0–4 (suboptimal), 5–7 (acceptable), 8 (optimal).

However, when K-S was applied to the differences in LVEF% (shown up at pre- vs. post-optimisation stage), the K-S test yielded a statistic of 0.218 with a p-value of 0.0021. Since p = 0.0021 < 0.05, the null hypothesis was rejected, indicating that the distribution of LVEF% differences was non-normal. In addition, given the longitudinal nature of the study, a *paired t-test* was subsequently used to assess statistical significance of the difference between baseline and post optimisation.

First method: z Critical Test. This study was hypothesis-driven, comparing a control group (pre-optimisation) to a study group (post-optimisation) of equal size and degrees of freedom. The null hypothesis ($H_0$) of our study posited that initiating and rapidly up-titrating GDMT to the maximum tolerated doses would not significantly impact LVEF and other key clinical indicators.

The analysis adopted the z critical value method based on relative frequencies.

A two-tailed test was conducted with an α = 0.05 (95% confidence level), allowing for both increases and decreases in values. The full statistical elaboration is available in Supplementary S2 Appendix.

Results indicated that the GDMT intervention was statistically significant for most clinical indicators. Despite this, the mean LVEF increased by 9.79%. Given the limitations of the z critical test in this scenario, further investigation was conducted using an alternative statistical method.

Second method: Paired t-Test. As established by the K-S test, the distribution of LVEF% differences was not normal, necessitating the use of a paired t-Test. Given the longitudinal design of the study, paired samples were analysed to test the null hypothesis of the up-titration when the patient is admitted into a virtual ward.

Key statistical equations.

Herein we briefly recall the relevant equations.

1. *Test Statistic Calculation*

$$SE = sd/\sqrt{n}$$

**sd** = Standard deviation of the differences, **n** = Number of pairs

$$t = d - \mu 0 / SE$$

**d⁻** = Mean of the differences 9, **μ0** = Hypothesized mean difference = 0

2. *Degrees of Freedom Calculation (minus 1)*

$$df = n - 1 = 86 - 1 = 85.$$

3. *Determination of p-Value*

The p-value was obtained using a t-distribution table or using an equivalent of statistical software. If p < 0.05, the null hypothesis was rejected.

Using this method, all measured difference variables led to the rejection of $H_0$, confirming that initiating and rapidly up-titrating GDMT to the maximum tolerated doses had a statistically significant effect on the clinical indicators analysed.

The full statistical results are provided in Supplementary S2 Appendix

## Ethical considerations

This work was conducted as a service improvement project rather than a research study. The project pathway was approved at the Emergency and Integrated Care (EIC) divisional quality board at Chelsea and Westminster NHS

Foundation Trust and by the London North West Virtual Ward Clinical and Operational Group. Consent was sought by all patients prior to enrolment in the rapid up-titration program.

## Results

Eighty-six patients were enrolled onto the heart failure virtual ward from April 2023 to August 2024. As illustrated in Tables 2, 3 and Fig 1, the patient descriptive characteristics showed an average age of 65 years (IQR 58-75), with 29% being female. The ethnic distribution was predominantly Caucasian (65%), followed by Asian (29%) and Black or African (7%). Most patients (92%) were admitted as admission avoidance, while the remaining 8% as part of early supported discharge (Fig 2). The highest incidence of comorbidities included ischaemic heart disease (33%), hypertension (33%), atrial fibrillation (29%), and diabetes (30%). Other notable comorbidities were high cholesterol (19%), asthma/COPD (21%), and chronic kidney disease (12%). A smaller proportion of patients (4.6%) had no past medical history (Fig 3).

### Primary outcomes

The median NYHA score at baseline was 2, indicative of moderate limitations in physical activity. The score improved to 1 at discharge post optimisation, reflecting minimal or no symptoms during routine activities ($p < 0.0001$). Similar improvements were seen in cardiac systolic function, as measured by the LVEF. The median LVEF increased from 30% (IQR 22-36) at baseline to 39% (IQR 31-49) post optimisation, signifying improved myocardial performance and a reduction in the severity of left ventricular dysfunction ($p < 0.0001$) (Figs 4, 5). The improvements in NYHA score and LVEF indicate that the intensive intervention effectively altered these parameters. The median time from diagnosis to initiation of

**Table 2. Patient demographics.**

| Characteristic | N = 86 |
|---|---|
| Age, mean, years (IQR) | 65 (58-75) |
| Female sex, n, (%) | 25 (29%) |
| Ethnicity, n (%) | |
| Caucasian | 56 (65%) |
| Asian | 25 (29%) |
| Black or African | 6 (7%) |
| Type of admission, n | |
| Admission avoidance | 79 (92%) |
| Early Supported Discharge | 7 (8%) |
| Comorbidities, n (%) | |
| Ischaemic Heart Disease | 28 (33%) |
| Hypertension | 28 (33%) |
| Atrial Fibrillation | 25 (29%) |
| Diabetes | 26 (30%) |
| High cholesterol | 16 (19%) |
| Asthma/COPD | 18 (21%) |
| Malignancy | 14 (16.3%) |
| Chronic Kidney Disease | 10 (12%) |
| Rheumatoid/Osteoarthritis | 9 (10.4%) |
| Peripheral Arterial Disease | 7 (8.1%) |
| Stroke/TIA | 5 (5.8%) |
| Deep Vein Thrombosis/Pulmonary Embolism | 5 (5.8%) |
| No Past Medical History | 4 (4.6%) |

optimisation was 4 days (IQR 2-8) and all patients were discharged after a maximum period of 4 weeks of monitoring and optimisation.

**Table 3. Age distribution.**

| | Min-Max | IQR | SD | Mode | Median | Mean | Variance |
|---|---|---|---|---|---|---|---|
| Age | 32-85 | 58-75 | 11.76 | 62 | 64.5 | 65.03 | 138.20 |

### DISTRIBUTION BY AGE

**Fig 1. Distribution by age.**

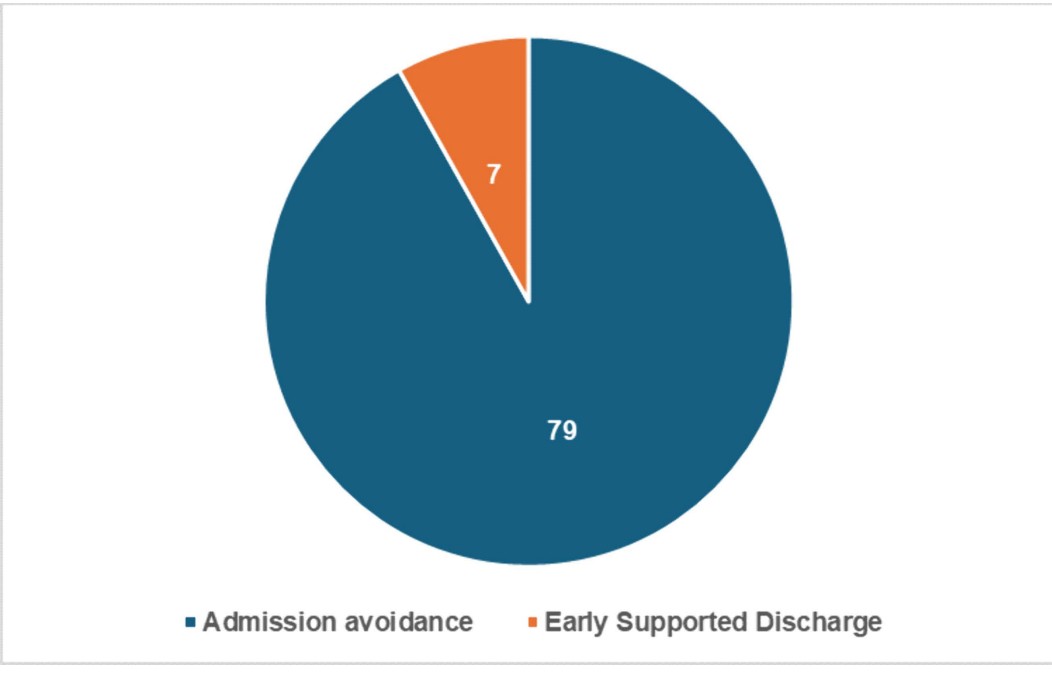

**Fig 2. Type of admission onto the heart failure virtual ward.**

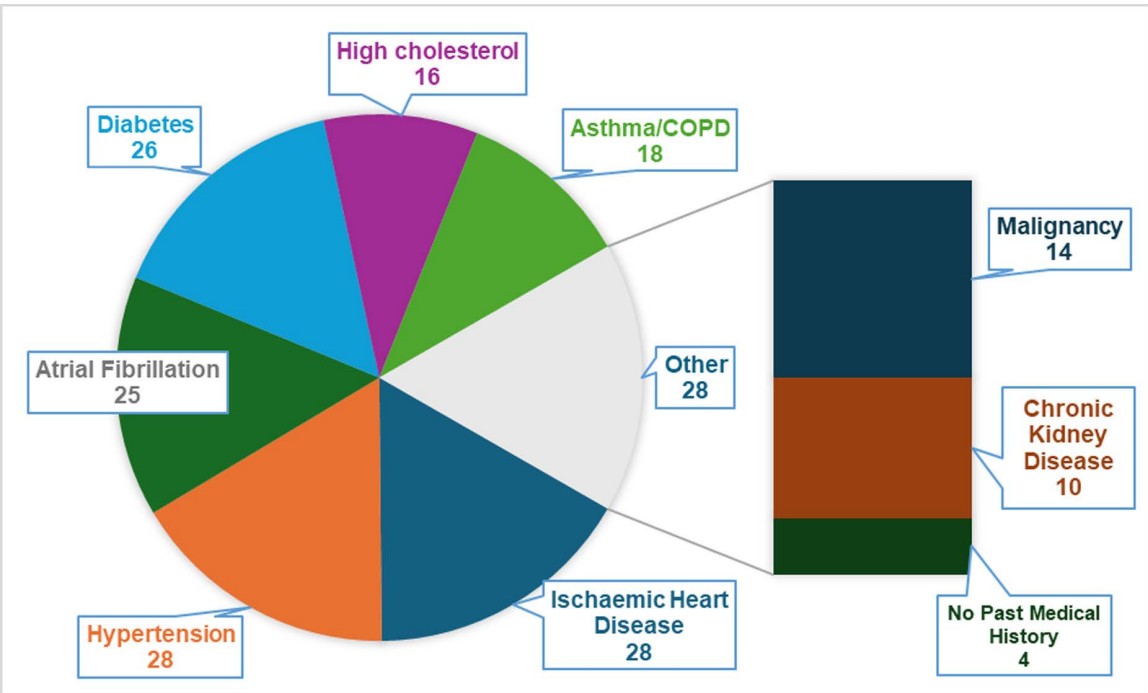

**Fig 3. Co-morbidities of patients undergoing rapid optimisation of GDMT.**

There was a significant improvement in the OMT scores (p < 0.001). At baseline, 81% of patients had a suboptimal OMT score (0–4), 19% had an acceptable score (5–7), with no patients achieving an optimal score of 8. Post-optimisation, 37% of patients attained an optimal score of 8, 52% attained an acceptable score (5–7), and only 5% remained in the suboptimal range (0–4) (Fig 6). The percentage of patients prescribed all four pillars of guideline-directed medical therapy increased dramatically from 8% at baseline to 84% post optimisation (Fig 7).

### Secondary outcomes

Over a 3-month follow-up period, hospitalisations related to heart failure were 8.1%, as evidenced by improvements in key clinical metrics such as NYHA class and LVEF, which are commonly associated with a lower risk of decompensation. Additionally, there was no notable increase in adverse events such as hypotension, bradycardia or hyperkalaemia. Blood pressure and heart rate values remained stable, with median heart rate, systolic and diastolic pressures reducing slightly but within clinically acceptable ranges (HR: 72bpm to 69bpm; SBP: 122–114 mmHg; DBP: 74–68 mmHg). Similarly, potassium levels showed a modest increase from a median of 4.4 mmol/L to 4.7 mmol/L, remaining well within the safe range. There was modest reduction in the mean eGFR from 85mL/min/1.73m$^2$ to 78mL/min/1.73m$^2$ which is accepted following the initiation of prognostic therapies. The European Society of Cardiology HF guidelines considers the following to be acceptable: an increase in serum creatinine of <50% above baseline, as long as it is < 266 umol/L, or a decrease in eGFR of <10% from baseline, as long as eGFR is > 25 mL/min/1.73m$^2$ [4]. Most patients experienced uneventful monitoring, with a total of 0.26 escalations being raised per patient over the optimisation period. These findings suggest that the optimisation process was not only effective in improving clinical outcomes but also safe and well-tolerated, with minimal need for escalated interventions during therapy titration. A summary of the main results is shown in Table 4.

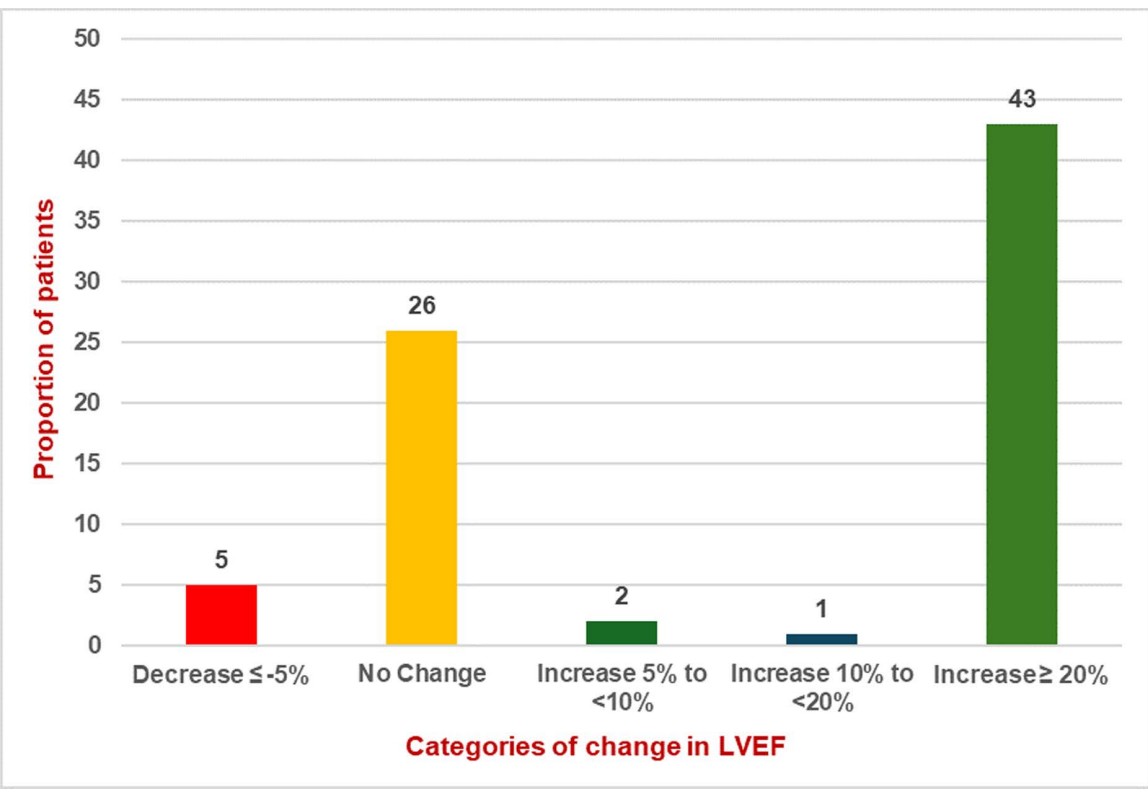

**Fig 4. Change in LVEF from baseline vs post optimisation. No change in LVEF was defined as a difference of <5% between the pre and post measurements.**

## Univariable analysis of clinical outcomes

To further explore the impact of the uptitration protocol, we conducted univariable analyses on three outcomes at 90 days: all-cause mortality, achievement of an OMT score of 8, and improvement in LVEF to greater than 35%. For each outcome, we report the odds ratio (OR), 95% confidence interval (CI), and associated p-value to estimate the effect size, precision, and statistical significance. These metrics provide insight into the plausibility and clinical relevance of the observed treatment effects, as shown in Table 5 below.

## Discussion

The rapid initiation and optimisation of GDMT in this study led to significant improvements in NYHA scores, LVEF, and OMT scores, reinforcing the positive impact on heart function, renal stability, and metabolic health. This aligns with existing evidence suggesting that early intervention with GDMT can halt or even reverse left ventricular remodelling, improve left ventricular function, and stabilise renal function, ultimately reducing the downstream risk of major cardiovascular events, including hospitalisations and mortality [12]. Several studies such as the CHAMP-HF and BIOSTAT-CHF registries have consistently shown that despite the proven benefits, there is an underutilisation of GDMT in heart failure management and delays in GDMT initiation leading to worse outcomes, including higher rates of heart failure hospitalisations and mortality [13,14]. Furthermore, Richard H. et al. demonstrated that the initiation of GDMT during hospitalisation was associated with improved survival rates, whereas discontinuation of therapy correlated with increased mortality [15].

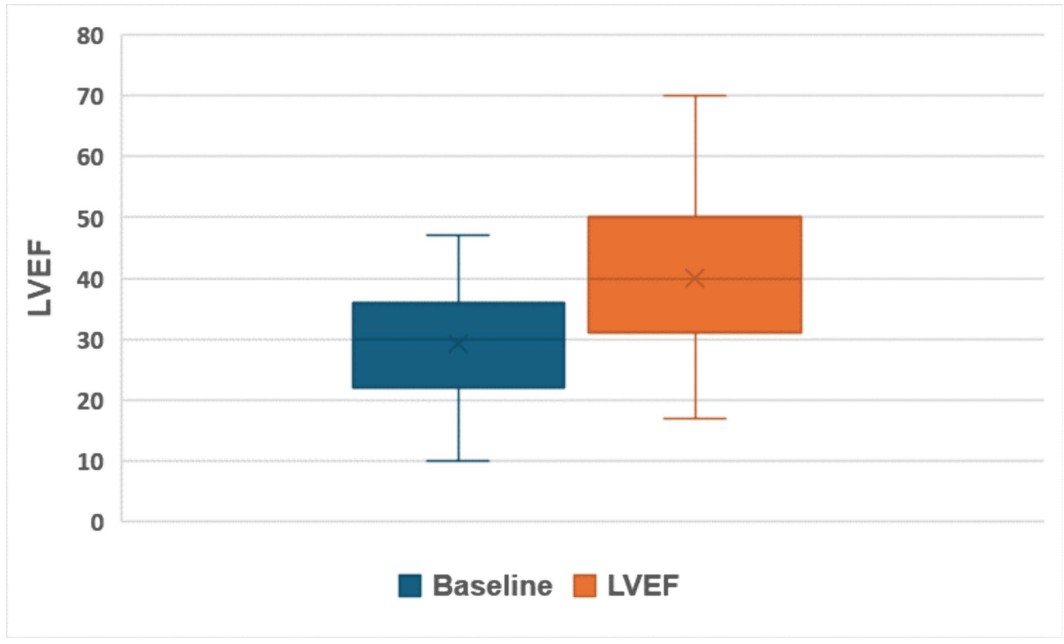

**Fig 5. The distribution of LVEF at baseline vs post optimisation with the x as the mean, the top and bottom of the box as the interquartile range and ends of whiskers as minimum and max recorded LVEF.**

Our study contributes to this growing body of evidence by also demonstrating that aggressive GDMT optimisation, even in a short timeframe, significantly reduces heart failure-related hospitalisation to 8.1%, compared to the UK's national 30-day readmission rate of 18% [16,17]. Notably, black ethnicity and chronic kidney disease (CKD) have been identified as key predictors of readmission, underscoring the importance of early and targeted interventions in these populations [16,17].

Our findings align with the STRONG-HF trial, which highlighted the importance of accelerated GDMT optimisation paired with close monitoring [8]. However, unlike STRONG-HF, which relied on in-person follow-up, our study leveraged a virtual ward model – a technology driven approached that enables remote care delivery while maintaining safety and continuity. Virtual wards have been increasingly recognised for their potential to improve chronic disease management by addressing common barriers such as clinical inertia, resource constraints, and patient accessibility issues. Recent studies on telehealth interventions in heart failure, including the TIM-HF2 trial, have demonstrated that remote monitoring can reduce all-cause mortality and heart failure hospitalisations [18]. The pharmacist-led model further enhances this strategy by ensuring evidence-based therapy adherence while mitigating workforce shortages in cardiology, aligning with NHS priorities on digital health integration, as emphasised in report by Darzi A [2].

Beyond clinical outcomes, this study highlights broader public health implications. Heart failure, CKD, and metabolic disorders such as diabetes mellitus form a tightly interlinked triad, often referred to as the cardio-renal-metabolic axis [19]. Dysfunction in one system can exacerbate the others through shared pathophysiological pathways such as inflammation, neurohormonal activation, and endothelial dysfunction. In HFrEF, renal impairment can worsen fluid overload and reduce the clearance of medications whilst metabolic disorders contribute to systemic inflammation, oxidative stress, and direct myocardial injury [19].

The high prevalence of T2DM (30%) and CKD (12%) in our cohort underscores the need for multidisciplinary strategies targeting these interconnected conditions. Studies such as DAPA-HF and EMPEROR-Reduced have

PLOS Digital Health

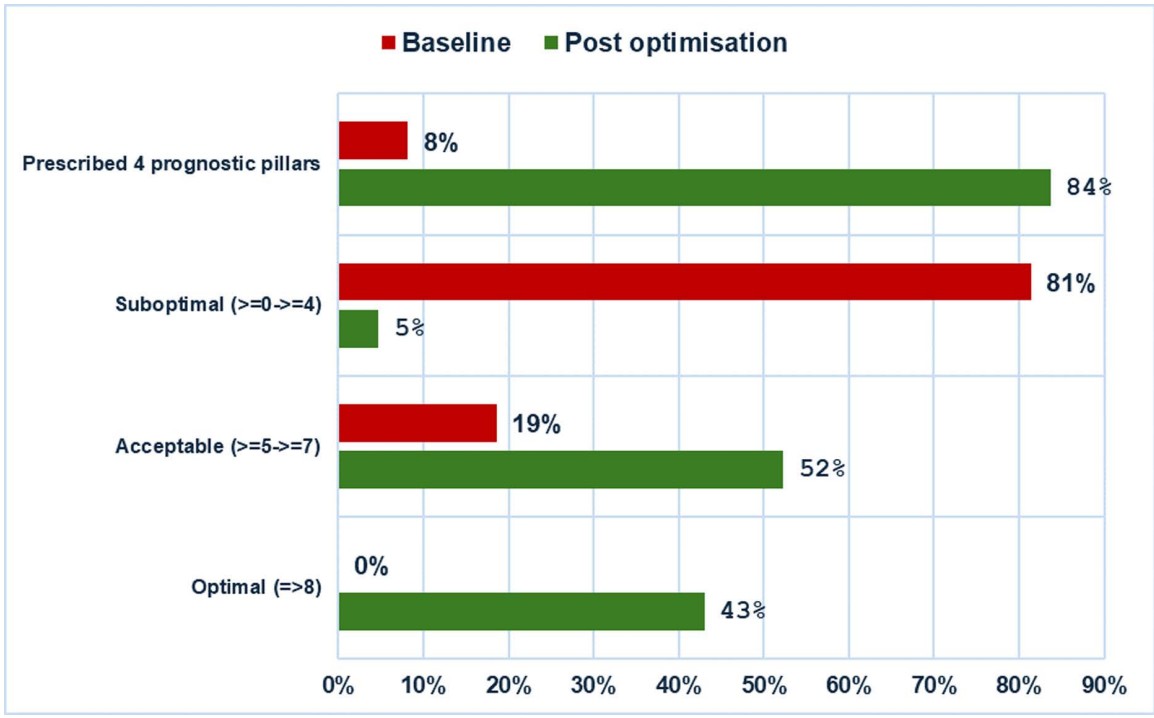

**Fig 6. OMT score at baseline and post optimisation and the % of patients prescribed the foundational four pillars of HFrEF on discharge from virtual ward.**

demonstrated that SGLT2 inhibitors not only improve cardiovascular outcomes but also provide renal and metabolic benefits, reinforcing the rationale for comprehensive therapeutic approaches [20,21]. The rapid initiation of GDMT in our study directly addresses this interconnection, promoting multi-system benefits that extend beyond cardiac function.

Despite its promising results, this study has limitations. The retrospective design and single-centre setting may limit generalisability, and while improvements in LVEF and NYHA scores are encouraging, the study's short follow-up period may not capture long-term outcomes. Evidence from large-scale studies suggests that LVEF improvement continues beyond 90 days; for instance, a study of 598 patients with de novo HFrEF reported LVEF improvement in 46% of patients at day 90, increasing to 68% at day 180 and 77% at day 360 [12]. This underscores the need for extended follow-up to fully capture the long-term trajectory of cardiac recovery and therapy benefits. The modest sample size and lack of subgroup analyses by demographics such as age, sex, and ethnicity also warrant caution in interpreting results.

Future research should focus on multicentre trials with larger, more diverse populations to validate these findings and explore scalability. Investigating the long-term sustainability of clinical benefits and cost-effectiveness of virtual ward models is also crucial. Subgroup analyses could provide insights into optimising care for specific patient populations, particularly those at higher risk of readmission. Additionally, the integration of artificial intelligence (AI) tools into remote monitoring hubs may further enhance the personalisation and efficiency of care delivery. AI-driven predictive analytics have shown promise in identifying early signs of decompensation in heart failure patients, potentially allowing for pre-emptive interventions.

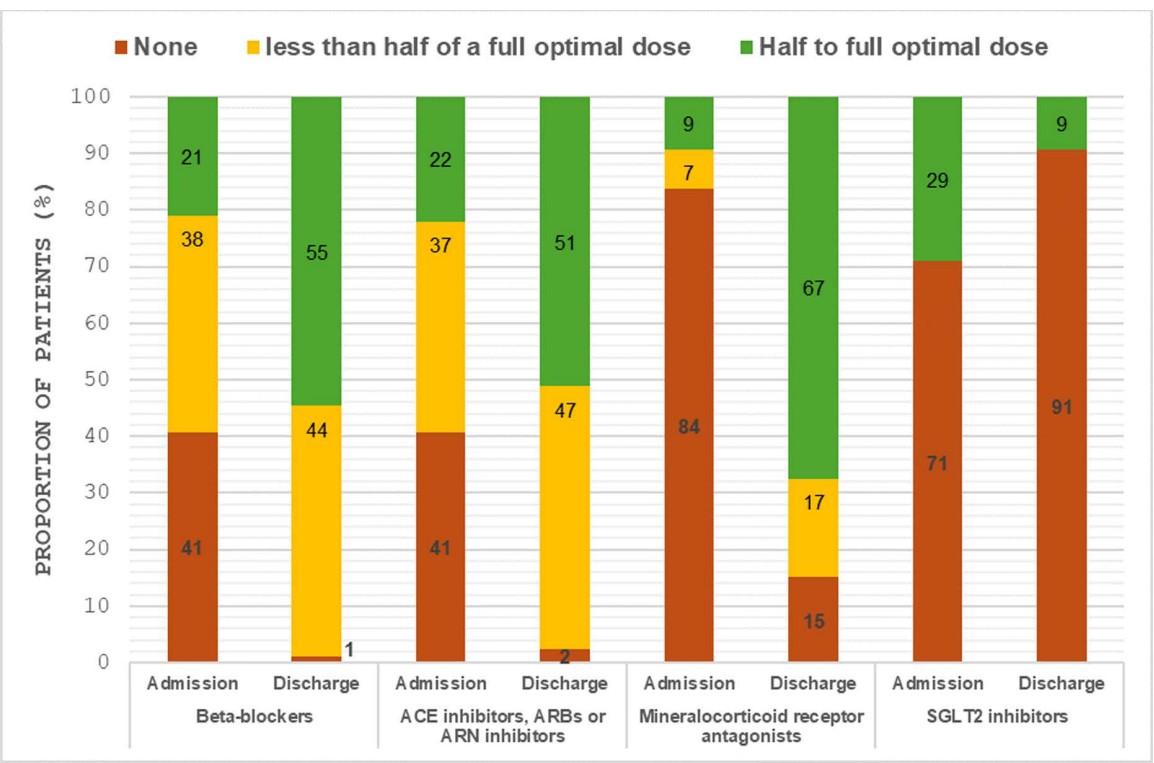

**Fig 7. Oral guideline-directed medical therapies for heart failure prescribed, on admission and on discharge from heart failure virtual ward.**

**Table 4. Results summary.**

| Parameter | Baseline | Post Optimisation | P value |
|---|---|---|---|
| NYHA, *median (n = 86)* | 2 | 1 | 0.0001 |
| LVEF (%), median, (IQR) (n = 86) | 30 (22-36) | 39 (31-49) | 0.0001 |
| Vitals/ laboratory values, median (IQR), (n = 86) | | | |
| HR, bpm | 72 (63-86) | 69 (59-74) | 0.0001 |
| SBP, mmHg | 122 (111-135) | 114 (104-123) | 0.0001 |
| DBP, mmHg | 74 (67-83) | 68 (61-77) | 0.0019 |
| K, mmol/L | 4.4 (4.1-4.6) | 4.7 (4.4-4.9) | 0.0001 |
| eGFR, *mean,* mL/min/1.73m$^2$ | 85 (72-90) | 78 (62-89) | 0.0001 |
| OMT score, median, (IQR) (n = 86) | 2 (0-4) | 7 (6-7) | 0.001 |
| Optimal (score 8) | 0% | 37% | |
| Acceptable (score 5–7) | 19% | 52% | |
| Suboptimal (score 0–4) | 81% | 5% | |
| Prescribed prognostic 4 pillars, % | 8% | 84% | |
| Heart failure admission within 3 months post optimisation, n (%) (n = 86) | | 7 (8.1%) | 0.01 |
| All cause death (n = 86) | | 5 (5.8%) | 0.012 |
| Time from diagnosis to starting optimisation, median, days (n = 86) | | 4(2-8) | |

**Table 5. Predictors of improvement of left ventricular ejection fractions >35%, OMT score and death at day 90.**

| | Univariable analysis | | |
| --- | --- | --- | --- |
| | OR | 95% CI | *P*-value |
| LVEF>35% | 1.35 | 0.88-2.07 | <0.001 |
| OMT score | 0.74 | 0.48-1.13 | < 0.001 |
| Death | 0.061 | 0.025-0.150 | <0.001 |

CI, confidence interval; OR, Odds Ratio; LVEF, left ventricular ejection fraction; OMT, optimal medical therapy.

## Conclusion

Remote up-titration of heart failure medications is a promising approach to overcoming clinical inertia, offering a fast, feasible, safe, and efficient treatment solution for patients who are otherwise undertreated. This study provides further evidence that GDMT is an effective approach for patients with HFrEF and can be provided safely within a virtual ward environment. Although our findings reached statistical significance, further investigation by adjusting the intervention timeline to allow for stabilisation of LVEF may offer additional benefits. Future work could increase the sample size, stratify by age, gender, and ethnicity to strengthen the results and improve generalisability. As NHS demand rises, amalgamating robust clinical practices with digital technologies has the potential to reduce admissions, improve efficiency, and deliver safe, impactful patient outcomes, which the present study highlights.

## Supporting information

**S1 Appendix. Raw data.**
(XLSX)

**S2 Appendix. Statistical analysis.**
(XLSX)

## Author contributions

**Conceptualization:** Sadia Khan.

**Data curation:** Hussein Alhakem, Angela Murphy, Sarah Pearse, Jodian Barrett.

**Formal analysis:** Hussein Alhakem, Liuba Fusco, Grant McQueen.

**Investigation:** Hussein Alhakem.

**Methodology:** Hussein Alhakem.

**Supervision:** Deirdre Linnard, Sadia Khan.

**Writing – original draft:** Hussein Alhakem.

**Writing – review & editing:** Hussein Alhakem, Angela Murphy, Grant McQueen, Sarah Pearse, Jodian Barrett, Deirdre Linnard, Sadia Khan.

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
