## [Decision Letter · Decision Letter 0]

21 Feb 2025

Response to Reviewers
Revised Manuscript with Track Changes
Manuscript
**Journal Requirements:**
**Additional Editor Comments (if provided):**
**Reviewers' Comments:**

**Comments to the Author**

1. Does this manuscript meet PLOS Digital Health’s publication criteria?

Reviewer #1: Yes

Reviewer #2: Partly

2. Has the statistical analysis been performed appropriately and rigorously?

Reviewer #1: Yes

Reviewer #2: No

3. Have the authors made all data underlying the findings in their manuscript fully available (please refer to the Data Availability Statement at the start of the manuscript PDF file)?

Reviewer #1: Yes

Reviewer #2: Yes

4. Is the manuscript presented in an intelligible fashion and written in standard English?

Reviewer #1: Yes

Reviewer #2: Yes

Reviewer #1: The paper covers an important topic of using remote patients monitoring for dose optimization and uptitration in heart failure patients with reduced ejection fraction. The topic is important because of the high prevalence of the disease and a high burden on healthcare system. The use of the virtual ward format for dose optimization is a reliable and convenient tool to reduce this burden while providing better and guidelines-based therapy for patients. The authors clearly show that this format of work helps to reduce NYHA scores and get better LVEF for patients by giving them optimized dioses of guideline-directed medical therapy. Despite the small number of patients and retrospective nature of the study it is worth publishing to drive attention of other colleagues to this effective and easy-to-implement approach. There are some questions to the statistical analysis of the data that should be addressed and clarified by the authors. Also please see below the list of comments worth correcting.

1. No references to tables and figures in the text – makes it more difficult to understand them and put them into the context. Please add.

2. The difference between new practice and standard practice need to be outlined more clearly.

3. P. 4, paragraph 2, last two lines: for non-English speaking readers there may be little difference between “seldom titrated” and “rarely titrated” – please, rephrase or give numbers.

4. P. 5, line 1: please insert “heart failure WITH reduced…”.

5. P. 9, line 3 from bottom: VW abbreviation not explained previously.

6. P. 11, line 3: please add point in the end.

7. P. 12, table, raw 3: please remove comma in “n (%)”.

8. Fig. 1: percentages are completely not the same as in the table 1; percentages of what number are shown? Possibly delete the figure at all, as the data are already shown in the Table 1.

9. Fig. 2: duplicating data from the Table 1 – possibly delete the figure?

10. P. 14, line 9: add “NYHA score”.

11. P. 14: as NYHA scores of 1 and 2 are quite close, it seems better to state the percentages of patients with different scores before and after titration (similar to OMT scores), with appropriate statistical analysis; to show the numbers of patients with better/same/worse scores.

12. P. 14, line 11-12: please add IQR or range of time from diagnosis to initiation of optimization and percentage of patients discharged after “most”, for those not discharged in time please specify reasons.

13. P.14: comparison of OMT scores – better to do with ANOVA or other tests for multiple groups.

14. P.14: for those already on inadequate doses of medications please specify the number (percentage).

15. Fig. 3: would be reasonable to have changes in LVEF in the gradual order: 1) decrease, 2) increase 5-<10%, 3) increase 10-<20%, 4) no change, 5) increase ≥20%.

16. P. 17, line 3; p.18, last line: p<0.02 compared to what?

17. Table 2, raw 4: alignment of numbers is necessary.

18. P. 19, paragraph 2, last line: possibly rephrase as “report by Darzi A.”.

19. P.19, paragraph 3: introduce data from the study a bit more into theoretical background.

20. P. 23, ref. 5: wrong reference format.

21. P. 23, ref. 8, line 4: remove comma before “trial”.

22. P. 24, ref. 15: remove semicolon before the name of the article; “association” and “classification” – check spelling; pages not stated.

23. Z-analysis mentioned in the Data analysis Supplement, but not in the text.

24. Not for all comparisons t-test works the best. Possibly it would be better to state statistical methods for all comparisons separately.

Reviewer #2: 1. The article does not follow high methodological rigor

2. Statistical analysis could be performed appropriately, but it is not presented properly

3. They attached some files, but not clear the utility and it is not even mentioned along the main article

4. Written well in english

**Do you want your identity to be public for this peer review?** For information about this choice, including consent withdrawal, please see our Privacy Policy

Reviewer #1: No

Reviewer #2: No

**Figure resubmission:****Reproducibility:** To enhance the reproducibility of your results, we recommend that authors of applicable studies deposit laboratory protocols in protocols.io, where a protocol can be assigned its own identifier (DOI) such that it can be cited independently in the future. Additionally, PLOS ONE offers an option to publish peer-reviewed clinical study protocols. Read more information on sharing protocols at https://plos.org/protocols?utm_medium=editorial-email&utm_source=authorletters&utm_campaign=protocols

---

## [Editor Report · Decision Letter 1]

28 Apr 2025

Pharmacist-led rapid uptitration clinic in heart failure patients with reduced ejection fraction: ourexperience within a virtual ward

PDIG-D-24-00601R1

Dear Mr Alhakem,

We are pleased to inform you that your manuscript 'Pharmacist-led rapid uptitration clinic in heart failure patients with reduced ejection fraction: ourexperience within a virtual ward' has been provisionally accepted for publication in PLOS Digital Health.

Best regards,

Francesco Andrea Causio

Guest Editor

PLOS Digital Health